# Low Bioerosion Rates on Inshore Turbid Reefs of Western Australia

Shannon Dee [1,*], Thomas DeCarlo [2], Ivan Lozić [3], Jake Nilsen [1] and Nicola K. Browne [1]

1 School of Molecular and Life Sciences, Curtin University, Bentley Campus, Bentley, WA 6102, Australia
2 College of Natural and Computational Sciences, Hawai'I Pacific University, 1 Aloha Tower Drive, Honolulu, HI 96813, USA
3 Centre for Microscopy, Characterisation, and Analysis, The University of Western Australia, Nedlands, WA 6009, Australia
* Correspondence: shannon.dee@bmtglobal.com

**Abstract:** Bioerosion on inshore reefs is expected to increase with global climate change reducing reef stability and accretionary potential. Most studies investigating bioerosion have focused on external grazers, such as parrotfish and urchins, whose biomass is more easily measured. Yet, cryptic endolithic bioeroders such as macroboring (worms, sponges and bivalves) and microboring taxa (fungus and algae) have the potential to be the dominant source of reef erosion, especially among inshore reef systems exposed to increased nutrient supply. We measured bioerosion rates of bioeroder functional groups (microborers, macroborers, and grazers), and their response to environmental parameters (temperature, light, turbidity, chlorophyll *a*), as well as habitat variables (coral cover, turfing algae, macroalgae) across two inshore turbid reefs of north Western Australia. Total bioerosion rates were low ($0.163 \pm 0.012 \, \text{kg m}^{-2} \, \text{year}^{-1}$) likely due to low light and nutrient levels. Macroborers were the dominant source of bioerosion and were positively correlated with turfing algae cover, highlighting the role of turf-grazing fish on endolithic bioerosion rates. Overall low bioerosion rates suggest that despite the reduced coral cover and carbonate production, these reefs may still maintain positive reef accretion rates, at least under current environmental conditions. However, an improved understanding of relationships between environmental drivers, habitat and grazing pressure with bioeroding communities is needed to improve predictions of reef carbonate loss with future climate change.

**Keywords:** macro-bioerosion; micro-bioerosion; grazers; microCT

## 1. Introduction

In recent years, there has been an increase in reef systems shifting to degraded states, whereby the destruction and erosion of reef framework outweighs reef accretion. One major source of erosion among reef systems is through bioerosion, which includes external grazing from taxa such as parrotfish and urchins, as well as endolithic boring and chemical dissolution from "microborers" (fungus or algae), and larger "macroborers" (sponges, worms, or bivalves; [1,2]). Bioerosion by endolithic borers has been shown to increase with eutrophication and warming waters [3,4]. These conditions also increase algal growth, which in turn results in more intensive grazing activity. Inshore turbid reefs are considered to be particularly vulnerable to higher rates of bioerosion as they are often exposed to elevated nutrients, and are situated in shallower, warmer waters [5]. These reef systems are likely to expand their geographic range in coming years due to sea level rise inundating coast lines, changing weather patterns (e.g., increased rainfall and river runoff) and continued modification of coastal catchments [6]. As such, we need to better understand how inshore reef systems are (and will be) impacted by increases in processes such as bioerosion that can destabilise reef systems.

The influence of bioerosion on reef function and development can be estimated by applying the carbonate budget technique (see review by Browne et al., 2021 [7]). A carbonate budget considers factors of reef accretion and erosion to calculate an estimated rate of net

reef accretion [8]. If a reef is considered to be in a negative budgetary state it is assumed to be degrading in terms of general coral health and structural complexity [9–11]. This has carry-on effects to other ecological reef functions such as loss of habitat and biological diversity [12]. Therefore, accurately estimating the rate of bioerosion is of increasing concern given that environmental drivers of bioerosion, particularly on inshore reefs, (e.g., eutrophication, warming oceans), are likely to increase in coming years.

Most carbonate budget studies have focused on external bioeroders whose abundance is easily measured from snapshot in situ observations. This has resulted in budgets that are heavily skewed to external grazing rates driven by the abundance of parrotfish and urchins along a transect [7]. Rates of bioerosion are then typically calculated based on offsite and/or historic relationships between these grazers, their size and bioerosion rates established in the 1980s and 1990s [13,14]. Although these studies provide empirical relationships between grazers and external bioerosion rates, snapshot observations of fish for carbonate budget assessments may overestimate or underestimate species abundance depending on season, recent local disturbance, as well as specimens fleeing the path of observing divers. Further, recent studies have shown that there may be significant spatial variation in parrotfish bioerosion activities across an individual reef platform [15,16], and that grazing may be more intense at specific times of the day, as well as between species [17]. Grazing can also intensify as a result of increased endolithic microborer and macroborer activity in response to environmental drivers (e.g., increased nutrients: [18–20]).

Comparatively, rates of endolithic bioerosion are limited, largely because it is more difficult to quantify compared to grazing rates [7] due to small size of the organisms, their patchy distribution (e.g., bivalves) and cryptic nature (e.g., sponges and polychaetes; [2,21–23]. To quantify these internal organisms, samples of live or dead corals have been collected for examination. The application of two-dimensional (2D) image analysis of coral cross-sections of live massive *Porites* provides an assessment of carbonate removed by bioerosion [24,25]. A less destructive method uses coral rubble, which is cut into cross-sections, and the volume of carbonate removed is determined using image analysis software [25–27]. This method does not require any expensive equipment and, as the rubble may have been in situ for many years, provides insight into the established bioeroder community and their relative abundances. However, this method cannot always provide reliable estimates of bioerosion rates as the length of time that the substrate has been available for bioeroding is ambiguous. Some studies have been able to 'time-stamp' rubble by assuming that a particular mortality event provided the rubble substrate [19,28,29] or by selecting pieces of rubble with limited algal growth and intact corallites to indicate that the rubble was produced <1 year previously [27].

An approach that has gained popularity in recent years uses pre-weighed *Porites* skeleton blocks deployed on to a reef (attached to the substrate) for 12 or more months [3,30–32]. This method is both less destructive (blocks are cut from cores) and is time-stamped. On removal from the reef, the block can be weighed to determine the mass of carbonate removed over time, and cut into cross sections to measure volume and identify boring organism (e.g., mollusc, sipunculan worm). More recently, experimental *Porites* blocks have been scanned using micro computerized tomography (microCT) before and after deployment [33–35]. The high-resolution scans obtained by microCT allow the user to analyse total volume loss and relate a percentage volume loss to macro and microboring. These methods likely provide some of the best estimates of endolithic bioerosion. However, the length of time of deployment of the blocks is important, with longer deployments (>1 year) typically allowing for ecological succession and the growth of slower growing eroders such as bivalves and sponges.

Spatial and temporal variations in bioerosion rates are driven by environmental and habitat differences. Given that most monitoring and research projects are unlikely to have the resources (time, funding) to either collect in situ samples or deploy blocks over several years, a more in depth understanding of what drives changes in the bioeroder community composition and rates of erosion may provide an alternative means of estimating changes

in reef erosion rates. Studies that have measured bioerosion rates with one or more environmental parameters have found strong relationships between bioerosion rates and environmental conditions, such as eutrophication [3,19,36], ocean acidification [35,37], and temperature [38]. However, there is a considerable lack of in situ data that have coupled bioerosion rates with changing environmental conditions, highlighting the need for studies that can provide data for the development of empirical relationships between key environmental parameters and rates of endolithic bioerosion.

Here, we apply the microCT method to measure rates of endolithic bioerosion of experimental *Porites* blocks deployed for one year at two inshore turbid reefs in Western Australia. Currently, there are no recorded rates of site specific endolithic bioerosion from West Australian reefs, with an additional paucity in bioerosion data from inshore reef systems globally. Site specific environmental (temperature, light, chlorophyll *a*, and turbidity) and habitat data were collected with micro-and macroboring rates. Estimates of fish grazing rates were also calculated from the experimental blocks and compared to grazing rates calculated from in-water fish census data to assess differences in rates between the two methods. Lastly, coral rubble was collected from the two reefs to determine if established endolithic communities (>1 year) were comparable to that observed in the experimental blocks. Together, these data provide new insights into rates of endolithic bioerosion on inshore turbid reefs, where high rates of bioerosion may push reefs from net accretion to net erosion. Further, by examining spatial differences in the bioerosion rates with environmental and habitat differences, we improve our understanding of drivers of bioerosion rates, and therefore, how bioerosion may fluctuate with future climate change.

## 2. Materials and Methods

### 2.1. Site Information

The Pilbara coast of north Western Australia hosts a number of well-developed inshore reef systems that are subject to frequent turbidity events due to large tidal ranges and the occasional storm surge [39–42]. This study was carried out at four sites on each of two inshore island reefs (Eva and Fly) located in Exmouth Gulf, which is situated at the southern end of the Pilbara coast (Figure 1). Eva reef ($-21.918454°$, $114.433502°$) and Fly reef ($-21.804829°$, $114.554003°$) have similar fringing reef morphology, coral cover (max cover = 63%, average cover = 8–10%), diversity (Shannon-Weiner index 0.73 and 0.76, respectively) and wave exposures (high exposure at northern reef sites, with little wave energy experienced at southern sites [43,44]). Coral carbonate production rates are $3.8 \pm 1.9$ kg m$^{-2}$ y$^{-1}$ at Eva reef and $2.9 \pm 1.5$ kg m$^{-2}$ y$^{-1}$ at Fly reef, which is low but typical for a turbid reef environment [43]. At each reef, sites included two northern wave exposed locations and two southern sheltered lagoon locations (Figure 1). Sites were referred to as being inshore or offshore in relation to the central island. The eight sites within this study were Eva south offshore (ESO), Eva south inshore (ESI), Eva north inshore (ENI), Eva north offshore (ENO), Fly south offshore (FSO), Fly south inshore (FSI), Fly north inshore (FNI), and Fly north offshore (FNI).

### 2.2. Environmental and Habitat Data

The study was conducted from April 2019 to May 2020. Temperature loggers (°C; Hobo Pendant UA-001-64) and photosynthetic active radiation (PAR) loggers (μmol photons m$^{-2}$ s$^{-1}$; Odyssey submersible PAR logger) were deployed at each site next to the blocks with logging intervals of 60 min for benthic temperature and 10 min for PAR loggers. Here, we only provide yearly average PAR and water temperatures values, which align with the yearly average bioerosion rates calculated. For a timeseries of the light and temperature data at these reef sites, please refer to Dee et al. [45]. Water quality variables of chlorophyll *a* (μg L$^{-1}$), salinity, pH and turbidity (FNU) were measured in situ monthly during neap tides at one site (southern offshore) per reef [45]. Sampling was undertaken using a vertical profiling method with a multi-parameter EXO Sonde 2 (YSI Inc./Xylem Inc.; details of testing methods outlined in Dee et al. [45]).

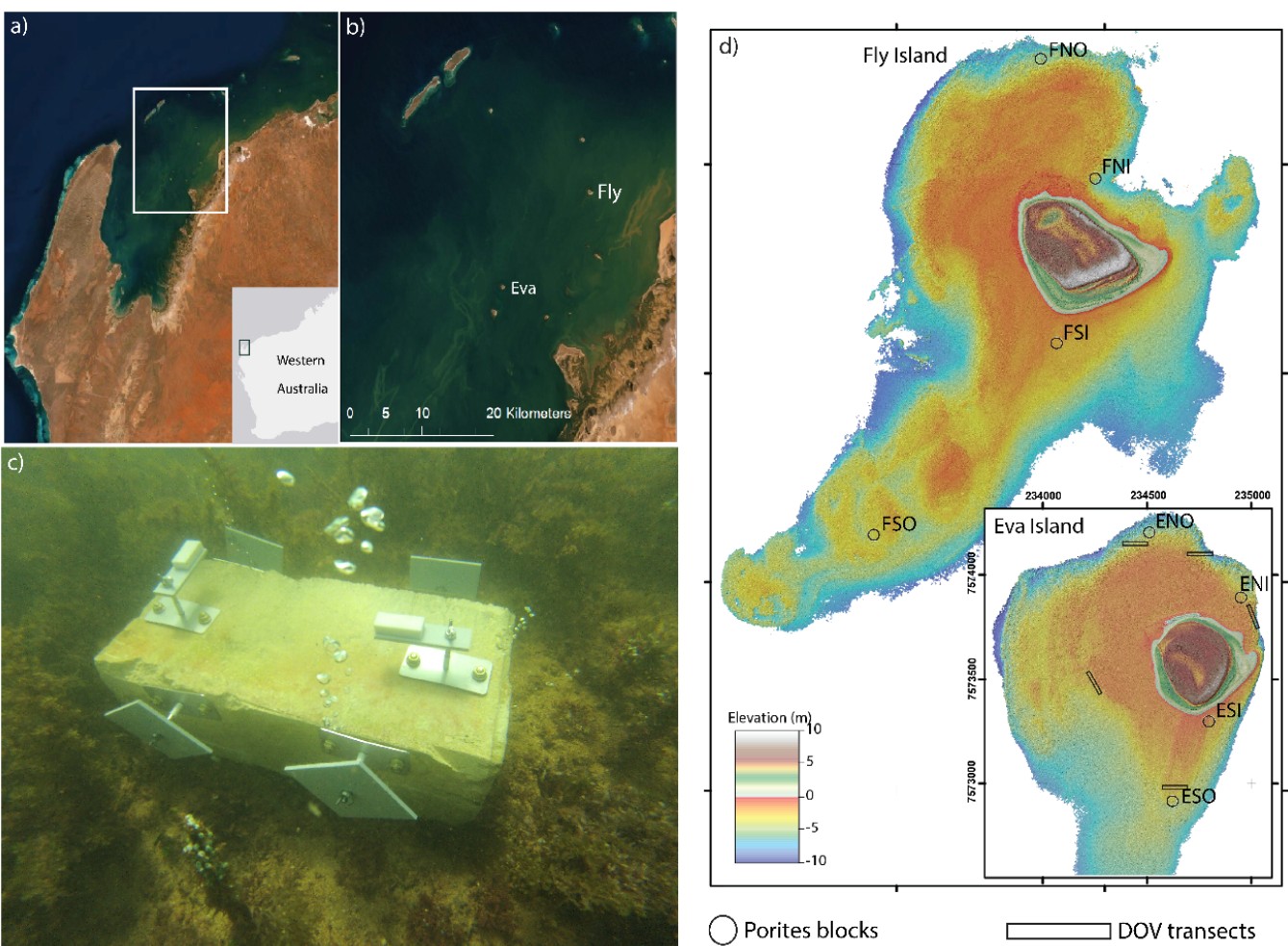

**Figure 1.** The West Australian coast shown in grey with a zoomed satellite image of Exmouth Gulf (**a**), highlighting the area around Eva and Fly Island reefs (**b**). Panel (**c**) shows one of two limestone blocks deployed at each of eight study sites, with the two 'bioerosion monitoring units' on top. Study sites are shown by black circles across Eva and Fly reefs in panel (**d**) on the right. The eight sites within this study were Eva south offshore (ESO), Eva south inshore (ESI), Eva north inshore (ENI), Eva north offshore (ENO), Fly south offshore (FSO), Fly south inshore (FSI), Fly north inshore (FNI), and Fly north offshore (FNI). Five 50 m long diver operated video (DOV) surveys were conducted around sections of Eva Reef to estimate the abundance and biomass of grazing fish. DOVs were conducted at ENI, ESO, ENO (×2), and the western zone.

Habitat data at each site were gathered by line intercept transects (20 m; see Dee et al. [43] for details). Briefly, along each transect, photos were taken every two seconds (<1 m apart) by a diver (approximately 0.5 m above seabed), totalling approximately 60 photos per transect. These photographs (capturing an area of approximately 2 m$^2$) were used to assess benthic habitat using Coral Point Count software (CPCe, [46]) where each photo was overlaid with 20 random points. Benthos under each point was classified into either coral, macro-algae, turfing algae or abiotic cover (e.g., sand, rock, rubble).

### 2.3. Micro CT of Porites Blocks

At each site, four blocks were deployed in early April 2019 and retrieved in mid-June 2020 (total number of blocks = 32). Each block (5 × 2 × 1 cm; average density = 1.56 g cm$^3$, Figure 2a) was obtained from cores of *Porites lutea* collected from the South Cay of Willis Island within the Coral Sea. Although cores were not collected at the study site, *Porites lutea* is the dominant massive coral at Eva and Fly reefs. Blocks were individually weighed

and attached to PVC plates (8 × 2 × 0.4 cm) with marine epoxy before being imaged using microCT at the NIF Bioimaging Facility located at the Centre for Microscopy, Characterisation and Analysis at the University of Western Australia, Perth, Australia. This attachment method has successfully been used in a number of recent bioerosion research studies, e.g., [35,47,48].

MicroCT scans were undertaken with a SkyScan 1176 microCT (Bruker-microCT, Kontich, Belgium) at 90 kV and 273 µA. Initial and post deployment scans were run at 35 µm resolution with a 0.1 mm Cu filter. Scans were reconstructed into image stacks using Bruker NRecon software using a modified Feldkamp cone-bream algorithm with ring artifact reduction of 20% and beam hardening of 20%. Pre- and post-deployment scans were directly compared through three-dimensional registration of the two data sets with Skyscan Data-Viewer software using the pseudo-3D registration strategy. From here we generated a bitmap "difference" image stack from the overlapped data sets, where eroded mass appeared white (bitmap value = 255), accretion appeared black (bitmap value = 0), and constant mass was grey. Three-dimensional analysis of this dataset in CTAn (version 1.18) allowed erosion to be measured by applying a threshold to isolate the white (bioeroded) regions. After the entire block was processed, a region of interest (ROI) was drawn for the interior area, excluding approximately 1 mm outer edge, to measure macroboring. This 1 mm exclusion was applied to measure microboring following evidence of microborers among inshore reefs of the Great Barrier Reef (Low Isles and Snapper Island), and other Indo-Pacific sites, boring to average depths of 1 mm over a 1-year period [17,49,50]. ROIs were also placed around obvious areas of scraping to measure external grazing. Rates of bioerosion (kg m$^{-2}$ year$^{-1}$) for each *P. lutea* block by grazers, microborers, and macroborers were measured as:

$$\text{Bioerosion rate} = (\text{Voli} \times \text{Di}) \div (\text{SAi} \times \text{time}) \tag{1}$$

where Voli is the volume of carbonate loss in the region of interest (areas of external grazing, internal and remaining outer 1 mm), Di is the density of the individual block of *P. lutea*, SAi is the surface area of the individual block, and time is the number of years the block was exposed (number of days/365; [33,37]. Identifications of macroborer borings, as seen in Figure 2c, were based on the characteristics of their borings following descriptions by Bromley [51] and Sammarco and Risk [24].

### 2.4. Coral Rubble

To assess how representative the macroboring community present in the experimental blocks of the long-term (>1 year) boring community was, we randomly collected 50 pieces of coral rubble from each reef. All samples collected were *Acropora* (~35%) or *Pocillopora* (~65%) branching species. After collection, rubble samples were soaked in 5% bleach for 24 hrs before being rinsed with fresh water and dried at 50 °C for 48 h. Samples were then cut into approximately 4 mm sections (three to five sections depending on rubble width) along the axis of vertical growth. Cross sections were photographed using a Canon EOS70D camera and used to determine the proportion of erosion due to the key macroboring organisms (bivalves, sponges, worms). Rates of bioerosion from coral rubble pieces were not calculated given that we could not accurately estimate the length of time the rubble pieces had been available for bioerosion. However, all rubble pieces collected had algal growth and eroded corallites, indicating that the coral had died several years previously and would, therefore, provide further insight into established boring communities.

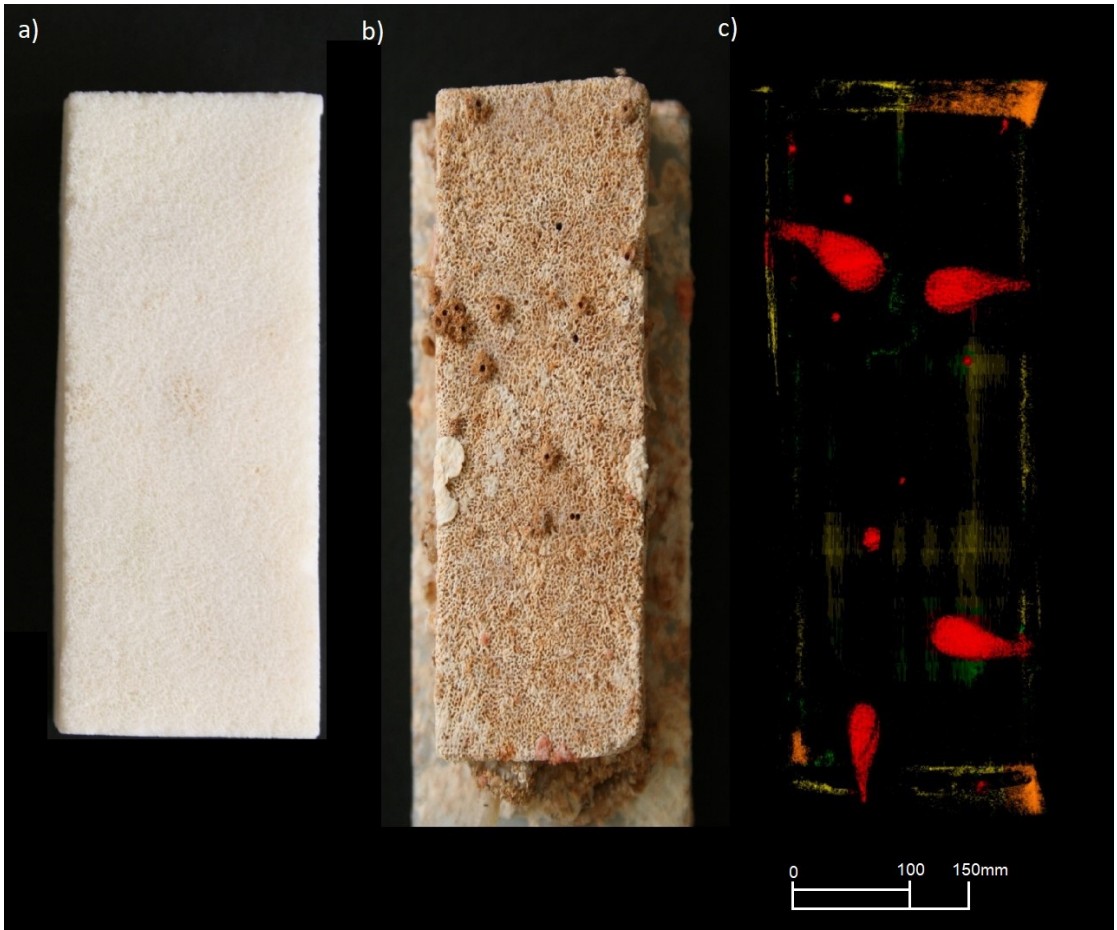

**Figure 2.** Experimental block of Porites lutea before (**a**) and after 12 months of deployment (**b**), along with micro computerized tomography (microCT) scan of the block post deployment (**c**). Varying sources of bioerosion displayed on the microCT image include grazing (orange), microboring (yellow), and macroboring by bivalves, polychaetes and sipunculans (red).

### 2.5. Fish and Urchin Grazing

To assess the suitability of estimated grazing rates determined from the experiment blocks, we also estimated grazing rates using visual fish census data. Diver-operated stereo-video (DOV) was used to collect data on fish abundance and biomass across Eva reef. Four 50 m transects separated by 10 m were carried out by a SCUBA buddy pair at five sites (Figure 1). Transects were kept linear and to a consistent depth profile (depth range across all sites = 3 to 4 m). The DOV system used two Sony ActionCams (FDR-X3000, Sony, Tokyo, Japan) mounted on either side of the base, approximately 800 mm apart, which were calibrated prior to fieldwork (for more information on the DOV system setup and its applications, see Goetze et al. [52]. Video footage was analysed using the computer software EventMeasure, which allows for accurate abundance and length measurements with a pre-populated species list [52]. All fish observed were measured to fork length and identified to their lowest possible taxonomic resolution. Count data were used where length measurements were not possible (i.e., due to limited visibility or obstruction of view). To ensure accurate identification and length measurements, values were excluded if their residual mean square (RMS) error exceeded 20 mm, and if measurement to length ratio precisions exceeded 10%. All measurements farther than 7 m from the DOV and outside of the 5 m wide belt transect were also excluded [52]. Biomass was calculated using fish fork length as a proxy for weight, in the equation:

$$\text{Weight (g)} = a \times \text{Length (cm)} \tag{2}$$

where *a* is the species-specific coefficient related to body form [53]. The coefficient is determined by plotting fish length and weight (i.e., regression). Coefficients at the family and genus level were used cautiously for species without published data, and for fish that were not identified to species level. Coefficient (slope) values and fish feeding guilds were derived from the FishBase website (https://www.fishbase.se/search.php accessed on 20 February 2021) and relevant published literature [54]. DOV's were not carried out across Fly reef due to limited days in the field, but as there is no significant variation in habitat types between Eva and Fly reef [43], and similar abundances have been witnessed at Fly reef previously (pers. obs.), we are confident that the abundance and biomass of grazing fish would be similar to that of Eva reef. Bioerosion rates for fish were determined using the "Indo-Pacific" data spreadsheet available from the *Reef Budget* website https://geography. exeter.ac.uk/reefbudget/indopacific/ (accessed on 12 March 2021); [55]. Urchin abundance estimates were conducted along each habitat line intercept transects at Eva and Fly reef. There were negligible amounts of urchins found across both reefs (<5 individuals per reef) and so they were excluded from analysis.

*2.6. Statistical Analysis*

To assess differences in rates of bioerosion between the three bioeroder functional groups (micro, macro, grazing) and between the two reef sites, a three-factor mixed-model nested ANOVA was conducted. Fixed factors were bioeroder functional groups and reef, and the random factor was site, nested in reef. The nested model was fitted using the lmer procedure (lmerTest Package version 3.1-3) [56]. The first model included the interactive effect between bioeroder functional groups and reef, which was found to not be significant, hence the model was re-run with no interactive effect. Where necessary, the Tukey Poct Hoc test was conducted using the multcomp package (version 1.4-2) [57].

A multiple linear regression was conducted separately for each bioeroder functional group to assess if rates of bioerosion were influenced by key environmental (temperature, light, chlorophyll *a*, turbidity, depth) and habitat variables (coral, macro-algae and turfing algae cover). Multi-collinearity between variables was checked using scatter plots and pairwise correlations. Light, chlorophyll *a* and turbidity were found to be highly correlated with other variables and were removed from the analysis. To further visualise the influence of environmental and habitat differences on bioerosion at the site level, we conducted a PCA analysis using the ggfortify package (version 0.4-1.4) [58]. Model assumptions of linearity, normality and homogeneity of variance were checked by graphically plotting model residuals. All data analysis were conducted in RStudio version 4.2.1 [59].

## 3. Results

*3.1. Environmental and Habitat Data*

Fly reef was characterised by higher levels of chlorophyll *a* (Fly mean = $0.49 \pm 0.06$ µg L$^{-1}$; Eva mean = $0.38 \pm 0.05$ µg L$^{-1}$), turbidity (Fly mean = $2.27 \pm 0.36$ FNU; Eva mean = $1.48 \pm 0.33$ FNU), and pH (Fly mean = $8.19 \pm 0.03$; Eva mean = $8.18 \pm 0.01$), while Eva recorded higher light levels (Fly mean = $177.52 \pm 1.73$ µmol photons m$^{-2}$ s$^{-1}$; Eva mean = $228.06 \pm 2.38$ µmol photons m$^{-2}$ s$^{-1}$) (Table 1). Habitat did not vary between reefs, but did vary between northern and southern sites, with northern sites typically supporting higher coral and turfing algal cover, and southern sites supporting macro-algae growth (Table 1).

**Table 1.** Average substrate (% cover) dominated by coral, macroalgae (MA), turfing algae (TA-mostly on dead coral), and sand. Mean annual environmental variables measured monthly throughout 2019/2020 at Eva and Fly reefs, and average ($\pm$ SD) microCT (microboring, macroboring, grazing, and total) (kg m$^{-2}$ year$^{-1}$) measured for each site.

| Reef | Site | Depth (m) | Coral | MA | TA | Sand | Light (PAR) | Temperature (°C) | Chlorophyll (µg L$^{-1}$) | pH | Turbidity (FNU) | Salinity | Micro | Macro | Grazing | Total |
|---|---|---|---|---|---|---|---|---|---|---|---|---|---|---|---|---|
| | | | | Substrate Cover (%) | | | | | Environment | | | | | Bioerosion Rates (kg m$^{-2}$ year$^{-1}$) | | |
| Eva | ENI | 3.1 | 9.51 | 56.44 | 19.02 | 13.50 | 142.04 (±73.15) | 27.15 (±0.80) | 0.38 (±0.05) | 8.18 (±0.01) | 1.48 (±0.33) | 38.16 (±0.42) | 0.03 (±0.01) | 0.06 (±0.01) | 0.01 (±0.01) | 0.11 (±0.02) |
| Eva | ENO | 3.5 | 65.35 | 0.00 | 13.52 | 5.07 | 142.04 (±59.90) | 27.15 (±0.80) | 0.38 (±0.05) | 8.18 (±0.01) | 1.48 (±0.33) | 38.16 (±0.42) | 0.03 (±0.01) | 0.10 (±0.02) | 0.01 (±0.01) | 0.15 (±0.07) |
| Eva | ESI | 2.7 | 2.06 | 62.06 | 3.82 | 19.41 | 311.17 (±73.15) | 26.90 (±0.86) | 0.38 (±0.05) | 8.18 (±0.01) | 1.48 (±0.33) | 38.16 (±0.42) | 0.07 (±0.01) | 0.06 (±0.01) | 0.02 (±0.01) | 0.16 (±0.02) |
| Eva | ESO | 3.6 | 1.19 | 71.85 | 5.12 | 7.77 | 311.17 (±59.90) | 26.90 (±0.86) | 0.38 (±0.05) | 8.18 (±0.01) | 1.48 (±0.33) | 38.16 (±0.42) | 0.07 (±0.02) | 0.10 (±0.05) | 0.02 (±0.01) | 0.22 (±0.02) |
| Fly | FNI | 4.5 | 29.12 | 0.00 | 34.71 | 20.29 | 106.62 (±36.10) | 27.20 (±0.89) | 0.49 (±0.06) | 8.19 (±0.03) | 2.27 (±0.36) | 38.34 (±0.46) | 0.05 (±0.01) | 0.06 (±0.01) | 0.06 (±0.09) | 0.18 (±0.1) |
| Fly | FNO | 4.0 | 6.24 | 29.46 | 32.10 | 7.07 | 106.62 (±36.10) | 27.20 (±0.89) | 0.49 (±0.06) | 8.19 (±0.03) | 2.27 (±0.36) | 38.34 (±0.46) | 0.04 (±0.01) | 0.09 (±0.03) | 0.01 (±0.003) | 0.14 (±0.03) |
| Fly | FSI | 3.0 | 1.06 | 35.98 | 4.76 | 35.71 | 127.08 (±65.53) | 27.32 (±0.89) | 0.49 (±0.06) | 8.19 (±0.03) | 2.27 (±0.36) | 38.34 (±0.46) | 0.04 (±0.00) | 0.11 (±0.05) | 0.02 (±0.007) | 0.17 (±0.05) |
| Fly | FSO | 3.1 | 0.00 | 67.58 | 1.21 | 21.06 | 127.08 (±65.53) | 27.32 (±0.89) | 0.49 (±0.06) | 8.19 (±0.03) | 2.27 (±0.36) | 38.34 (±0.46) | 0.05 (±0.01) | 0.21 (±0.02) | 0.04 (±0.01) | 0.30 (±0.02) |

### 3.2. Bioerosion

Average total bioerosion rates of the experimental blocks was higher at Fly reef ($0.175 \pm 0.02$ kg m$^{-2}$ year$^{-1}$) than at Eva Reef ($0.151 \pm 0.01$ kg m$^{-2}$ year$^{-1}$), although reef differences were not significant (Table 2). Both reefs displayed similar levels of bioerosion within functional groups, with macroboring being the most dominant across both reefs (Figure 3). Boring rates between the three different eroder functional groups were significantly different, with highest rates from macroboring ($0.091 + 0.01$ kg m$^{-2}$ year$^{-1}$), followed by microboring ($0.048 + 0.01$ kg m$^{-2}$ year$^{-1}$) and grazing ($0.025 + 0.01$ kg m$^{-2}$ year$^{-1}$; Figure 3, Table 2).

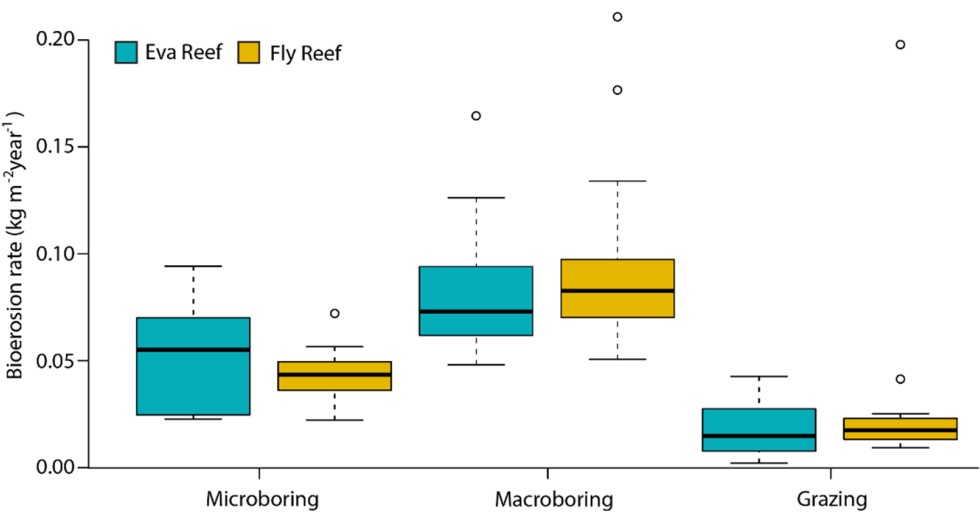

**Figure 3.** Bioerosion rates of functional groups (micro, macro and grazing) at Eva and Fly reefs, with outliers displayed as circles.

The bioeroder community of the *Porites* blocks and coral rubble samples were similar. Proportionally, the number of individual borings visible in the *Porites* blocks were approximately 85% worm (polychaete and sipunculan), and 14% bivalve borings. In the coral rubble samples, the proportional number of visible borings was 77% worm, 17% bivalve borings, and 6% sponge (Figure 4). It is acknowledged, however, that some cavities appearing to be worm borings within rubble samples, may in fact be the result of sessile worm species that had recruited to the exterior of corals when they were living. Sessile worms (Serpulidae and Sabellidae) form an exterior calcareous tube, which can be overgrown by the live coral and remain within the coral skeleton, creating cavities that are not the product of bioerosion [60–62].

**Table 2.** Three factor mixed model nested ANOVA with bioeroder functional group and reef as fixed factors and site as a random factor nested in site. Post hoc Tukey conducted for bioeroder functional group.

| Fixed Effects | Sum Squares | Mean of Squares | DF | F Value | *p* Value |
|---|---|---|---|---|---|
| Bioeroder | 0.060 | 0.030 | 2 | 28.00 | **<0.001** |
| Reef | 0.001 | 0.001 | 1 | 0.89 | 0.428 |
| **Random effects** | **Variance** | **SD** | | | |
| Reef:Site | 0.00003 | 0.005 | | | |
| Site | 0.00002 | 0.004 | | | |
| Residual | 0.00108 | 0.032 | | | |
| **Post hoc Tukey** | **Estimate** | **SE** | **Z value** | *p* value | |
| Macro-Micro | 0.043 | 0.009 | 4.80 | **<0.001** | |
| Grazing-Micro | −0.023 | 0.009 | −2.58 | **0.027** | |
| Grazing-Macro | −0.066 | 0.009 | −7.38 | **<0.001** | |

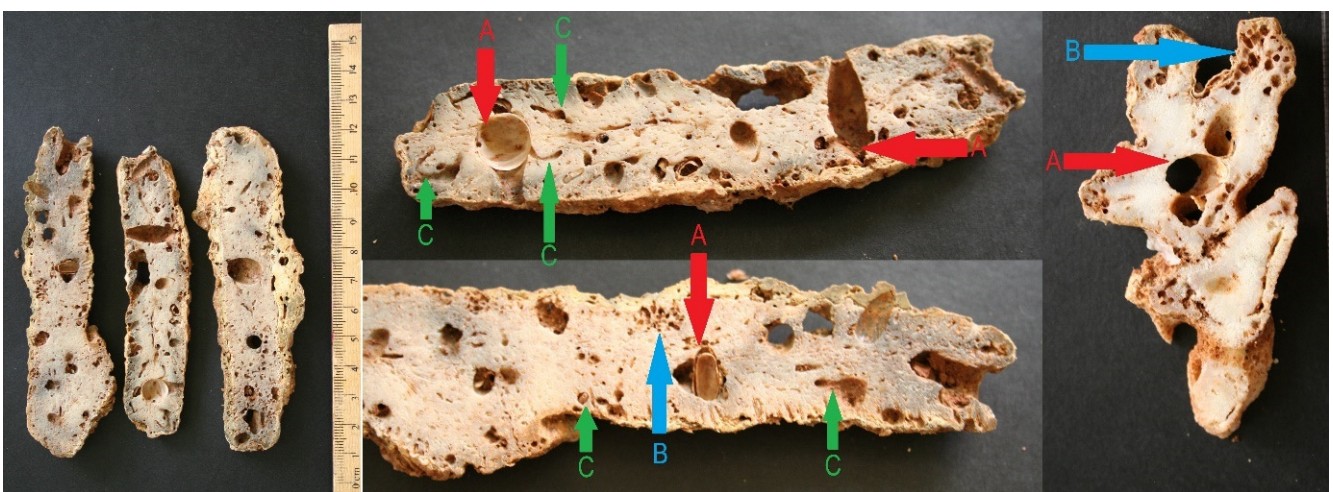

**Figure 4.** Branching coral cross sections from Eva and Fly reefs showing endolithic boring by established (A) bivalve, (B) sponge, and (C) worm species.

### 3.3. Fish Abundance and Erosion Rates

DOV data found that the fish community was largely composed of omnivores ($40 \pm 15$ per 1000 m$^2$), carnivores ($21 \pm 19$ per 1000 m$^2$) and invertivores ($19 \pm 5$ per 1000 m$^2$) across Eva. In contrast there was low abundance of herbivorous fish species ($7.5 + 7$ per 1000 m$^2$), with the only parrotfish species (*Scarus ghobban*) having an average density of $2.6 \pm 0.75$ per 1000 m$^2$. Average biomass of *Scarus ghobban* was $15.53 \pm 7.33$ g per km$^2$ resulting in an estimated average bioerosion rate of $0.059 \pm 0.030$ kg m$^{-2}$ year$^{-1}$ (Figure 5). This is approximately 3.5 times the average grazing rate recorded by *Porites* blocks at Eva ($0.017 \pm 0.003$ kg m$^{-2}$ year$^{-1}$).

### 3.4. Environmental and Habitat Drivers of Site Differences in Bioerosion

The PCA biplot shows a clear separation of sites along both the *X* axis (56.2% of the variance) and the *Y* axis (21.7% of the variance; Figure 6). Here, we have further grouped the eight sites into bioeroder community groups dominated by grazing, macroborers, micro-borers or mixed. High light and macro-algal cover as well as low temperature was more closely associated with micro-borer dominated communities (sites ESI and ESO) whereas the site where grazing was high (site FNI) was more closely associated with higher temperature, low light and high turfing algal cover. Those sites where macroborers were the dominant bioeroder group (ENO, FNO) were those that had higher levels of turfing algal cover.

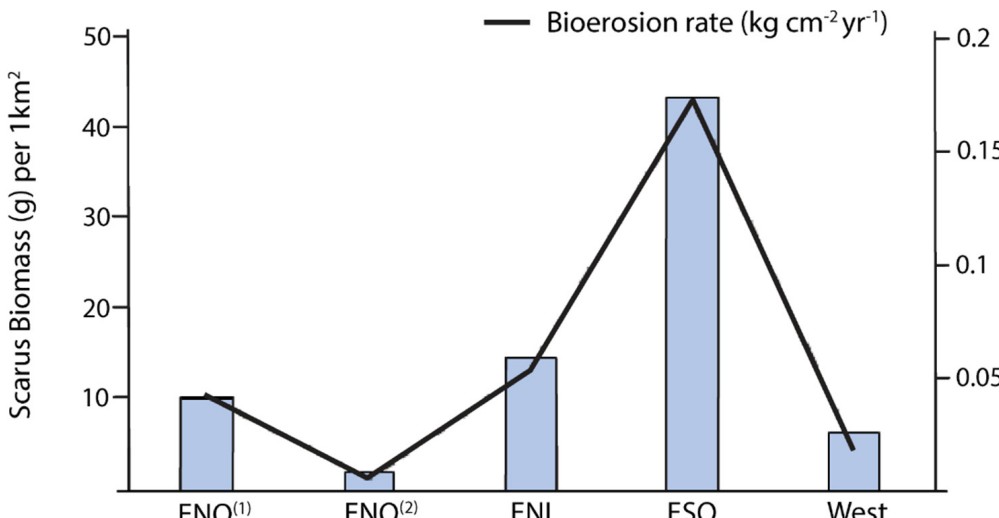

**Figure 5.** Estimated biomass of grazing herbivore *Scarus ghobban* across five DOV transects and the estimated bioerosion rate (kg m$^{-2}$ year$^{-1}$) calculated using the *Reef Budget* data sheet for the Indo-Pacific.

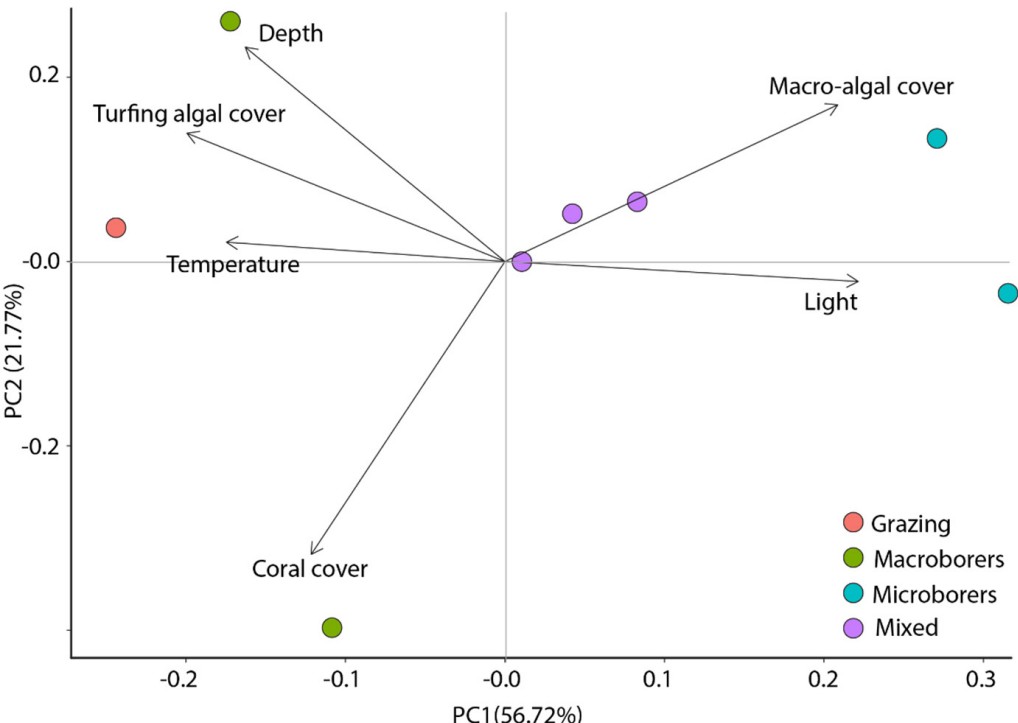

**Figure 6.** Principal component analysis (PCA) of environmental (temperature, light, depth) and habitat variables (coral, macro-algal and turfing algal cover) from all eight sites. Here, sites have been grouped according to their dominant bioeroder group (macroborers, microborers, grazer or mixed).

Multiple linear regression of endolithic bioerosion rates measured using *Porites* blocks with environmental variables measured during the time of deployment showed varying results. The model for microboring explained 27% of the variation in spatial micro-bioerosion rates ($p = 0.037$) with temperature having a significant and negative effect ($p = 0.008$; Table 3). The model for macroborers explained 29% of the variation ($p = 0.029$) with depth having a negative effect ($p = 0.006$) and turfing algal cover having a positive effect. The model for grazers was not significant as were the variables included in the model.

**Table 3.** Multiple linear regression of endolithic bioerosion rates (micro, macro, and total) measured using Porites blocks with environmental and habitat variables measured during the time of block deployment.

| Microborers | Estimate | SE | T Value | *p* Value | Multiple $R^2$ | Model Adjusted $R^2$ | F Statistics | *p* Value |
|---|---|---|---|---|---|---|---|---|
| Temperature | −0.0949 | 0.8969 | 2.948 | 0.008 | | | | |
| Depth | 0.0058 | 0.0122 | 0.472 | 0.642 | | | | |
| Coral cover | −0.0006 | 0.0004 | −1.589 | 0.127 | 0.411 | 0.270 | 2.926 | **0.037** |
| Macroalgal cover | −0.0005 | 0.0004 | −1.476 | 0.155 | | | | |
| Turfing algal cover | −0.0008 | 0.0006 | −1.446 | 0.163 | | | | |
| **Macroborers** | | | | | | | | |
| Temperature | −0.0527 | 0.0620 | −0.851 | 0.404 | | | | |
| Depth | −0.0720 | 0.0236 | −3.055 | **0.006** | | | | |
| Coral cover | −0.0005 | 0.0007 | −0.682 | 0.503 | 0.426 | 0.290 | 3.124 | **0.029** |
| Macroalgal cover | 0.0003 | 0.0007 | 0.441 | 0.664 | | | | |
| Turfing algal cover | 0.0042 | 0.0011 | 3.739 | **0.001** | | | | |
| **Grazers** | | | | | | | | |
| Temperature | 0.0161 | 0.0201 | 0.798 | 0.435 | | | | |
| Depth | −0.0086 | 0.0076 | −1.132 | 0.271 | | | | |
| Coral cover | −0.0001 | 0.0002 | −0.202 | 0.842 | 0.105 | −0.118 | 0.4716 | 0.793 |
| Macroalgal cover | 0.0001 | 0.0002 | 0.173 | 0.865 | | | | |
| Turfing algal cover | 0.0003 | 0.0004 | 0.932 | 0.362 | | | | |

## 4. Discussion

This study provides the first rates of bioerosion for marginal reef systems in Western Australia, filling an existing data gap in bioerosion rates on marginal reefs. Total bioerosion rates (average 0.16 kg m$^{-2}$ year$^{-1}$) were comparable to other studies on inshore turbid reefs of the Great Barrier Reef (GBR), and to studies adopting microCT analysis. For example, Tribollet and Golubic [63], assessed rates of bioerosion at two inshore island reefs (Snapper Island and Low Isles) of the GBR using 2D image analysis of experimental *Porites* blocks. These sites have similar fringing reef structure and turbidity levels to Eva and Fly, and recorded total bioerosion rates of 0.27 kg m$^{-2}$ year$^{-1}$ at Snapper Island and 0.18 kg m$^{-2}$ year$^{-1}$ at Low Isles following 1 year of deployment. Silbiger et al. [33,64] used microCT of experimental *Porites* sp. blocks and also found similar bioerosion rates ranging between 0.072 and 0.15 kg m$^{-2}$ year$^{-1}$ at sites in Hawai'i. Yet, these bioerosion rates are low compared to many other studies such as Sammarco and Risk [24] who measured bioerosion rates of between 1.21 and 11.13 kg m$^{-2}$ year$^{-1}$ on the GBR using x-rays of *Porites* heads, and Mallela and Perry [29] who recorded bioerosion rates ranged from 0.33 to 2.6 kg m$^{-2}$ year$^{-1}$ on an inshore reef in the Caribbean using coral rubble. Bioerosion rates have been found to be lower on inshore turbid reefs compared to offshore clearwater reefs, e.g., [49], likely due to lower light levels, which can reduce rates of micro-boring [64], as well as low scarid abundance and hence reduced grazing. Inshore reefs are, however, more vulnerable to increases in nutrients from terrestrial sources, which increases rates of bioerosion [20,51]. Although light levels are reduced in Exmouth Gulf due to elevated turbidity, nutrient inputs are low due to the aridity of the region [65] (chlorophyll *a* measured here was <0.49 µg L$^{-1}$). Hence, the combination of low light, scarid abundance, and nutrient levels has likely resulted in low total bioerosion rates at Eva and Fly reefs.

It could be argued that low bioerosion rates are potentially due to the short deployment of the experimental blocks (1 year). Many studies have found that as the length of deployment of experimental substrates increases, the rate of bioerosion increases [18,19,66]. This is due to ecological succession with microborers inhabiting dead coral substrate rapidly, followed by short-lived polychaete species [67]. Kiene and Hutchings [31] suggest that it is not until three years of exposure that bivalves and sponges populate available substrate. Macroborers typically have higher rates of internal bioerosion than micro-borers due to their size and activity, and if they are not recruiting to available substrate within the first year,

then bioerosion rates will be underestimated. Macro-bioerosion rates at Eva and Fly reef were comparable to previous studies over 1-year deployments, e.g., [63], but were typically lower than those conducted over several years. Our comparison of the endolithic bioeroder community between the experimental blocks and coral rubble found similar proportion of bivalves. This suggests that the rate of recruitment of bivalves is faster at inshore regions compared to offshore clearwater regions, as was previously observed on the GBR [63]. Recent research at Eva Reef on sediment composition found that sediments were dominated by both mollusc and coral fragments [68]. In addition, here we saw a greater abundance of fish invertivores ($19 \pm 5$ per 1000 m$^2$) than herbivores ($2.6 \pm 0.75$ per 1000 m$^2$). Taken together, these data suggest that there is a healthy mollusc population on these reefs, which has resulted in the rapid colonisation of available substrate.

Unlike macro-bioerosion rates, micro-bioerosion rates following one year typically represent longer-term bioerosion rates. Here, we measured micro-bioerosion rates of $0.051 \pm 0.007$ and $0.044 \pm 0.003$ kg m$^{-2}$ year$^{-1}$ at Eva and Fly reefs, respectively. Similar rates were observed by Chazottes et al. [3] in French Polynesia (0.044 to 0.067 kg m$^{-2}$ year$^{-1}$). However, these rates are typically lower than previously measured using experimental substrates (e.g., 0.2 to 0.57 kg m$^{-2}$ year$^{-1}$ [17] and 0.22 to 0.24 kg m$^{-2}$ year$^{-1}$ [69]). Lower micro-bioerosion rates on inshore turbid reefs have previously been attributed to low light as the result of suspended sediments in the water column and sediment entrapment in the epilithic algal turf [63], as well as high urchin grazing pressures, e.g., [3]. Grazers can remove the outer layer of the substrate where micro-borers are most active, with urchins being more efficient grazers than fish [70,71]. Given that grazing pressure is low on Eva and Fly reef, low micro-bioerosion rates are most likely low due to limited light as opposed to the influence of grazers. However, it is also important to note that microboring cavities range from 1 μm [72] up to 100 μm, and thus our microCT resolution of 35 μm may lead to underestimates of the total micro-boring rates.

Low grazing rates are typical of turbid inshore reefs characterised by low populations of parrotfish and urchins, e.g., [73]. Consistent with this expectation, external erosion on experimental blocks at Eva and Fly reefs was relatively low (average = 0.017 kg m$^{-2}$ year$^{-1}$ at Eva and 0.033 kg m$^{-2}$ year$^{-1}$ at Fly). We interpret the external erosion observed on the blocks as results of herbivorous fish, such as parrotfish, given that urchin numbers were very low on both reefs. The estimated average bioerosion rate from our DOV surveys was 0.06 kg m$^2$ year$^{-1}$, which was 3.5 times the average rate of external erosion captured on experimental blocks. Although DOV surveys cover a greater area of the reef, they are a snap-shot assessment of grazing pressure on reefs compared to the experimental blocks, which are deployed for an extended period. Additionally, the DOV approach relies on the extrapolation of observed relationships between fish length and bite rates, making it challenging to identify which estimate is more accurate. Regardless, bioerosion rates from both approaches were considerably lower than typically reported on coral reefs. For example, grazer bioerosion rates measured using fish data on the mid-shelf GBR reefs can range from 5.2 to 8.4 kg m$^2$ year$^{-1}$ [74] and a more recent study in the Central Indian Ocean found rates of 3.1 to 4.5 kg m$^2$ year$^{-1}$ [75]. Grazing rates using experimental substrates are consistently lower than fish survey estimates (e.g., 1.27 to 2.49 kg m$^2$ year$^{-1}$ [22]; 0.02 to 0.85 kg m$^2$ year$^{-1}$ [66]), but are typically greater than that measured here. However, of those studies that did measure grazing bioerosion rates on inshore turbid reefs using experimental substrates (e.g., Tribollet and Golubic [63]; 0.004 to 0.01 kg m$^2$ year$^{-1}$), the estimated rates were comparable to our observations in the Exmouth Gulf. Importantly, the only parrotfish observed was *scarus ghobban*, which is a scraper, as opposed to an excavator [76], and scrapers typically have lower bioerosion rates. Overall, our data suggest that grazing bioerosion rates in the Gulf are relatively low compared to other reefs around the world.

Spatial differences in micro-bioerosion are driven by environmental and habitat differences. Here, micro-bioerosion was negatively associated with temperature. Conversely, previous studies have found rates of micro-bioerosion are enhanced by increasing temper-

atures. For example, Reynes-Nivia et al. [77] found that micro-bioerosion of dead *Porites* skeleton roughly doubled when temperature increased from 24 °C to 28 °C together with an increase in $pCO_2$ from 400 μatm to 610 μatm, respectively. This same study also demonstrated the importance of light on skeletal dissolution rates, which are predominately driven by photosynthetic microborers. In the Exmouth Gulf, the negative effects of temperature on microboring were potentially due to temperature anomalies (up to 3.6 °C) recorded during the summer months, which may have caused thermal stress and decreased microboring activity to a greater degree at warmer sites. In addition, we also observed that as light levels increased, rates of micro-bioerosion rates increased (see Figure 6). However, due to the multi-collinearity of light with several other drivers, it was removed from the regression analysis. PCA analysis also indicated that those sites with higher micro-bioerosion rates were characterised by a higher macro-algal cover. This has previously been observed on the reefs of Reunion Island, Indian Ocean, where micro-boring rates were greatest at sites that were nutrient enriched and were associated with high macroalgal cover [4]. This association between micro-borers and the macroalgal cover is potentially related to reduced grazing pressure. When the macroalgal cover is very high, grazing pressure falls [77] and, as such, the outer layers of the substrate where the microborers reside are not removed. Given that grazing pressure was extremely low at all sites, the more likely explanation is that higher light levels in shallow waters are supporting both microborers and macroalgal cover.

In contrast, macro-bioerosion was positively correlated with turfing algal cover. A similar relationship between macroboring worms and algal turf was observed in Kenya [19], where reefs displaying higher worm abundance were also characterised by lower scarid fish abundance (due to fishing), denser algal turf and higher sea urchin abundance. In contrast, in those reefs with high scarid abundance (protected areas) and less dense algal turf, the macroboring community was dominated by sponges. In the Exmouth Gulf, worms dominated the endolithic community in both the experimental blocks and coral rubble, whereas there was either no or limited observation of sponges. Previous studies have also demonstrated that algal turf communities maintained by damselfish favour increased rates of internal bioerosion [78] as well as reduced rates of external grazing by parrotfish and urchins [79]. Damselfishes are common on inshore turbid reefs, including Exmouth Gulf where they represented 44% of fish recorded in the DOV. These fish will maintain patches of algal turfs and therefore, indirectly play a role in the macro-bioerosion rates on these reef types.

In summary, we found lower-than-expected rates of bioerosion compared to average global rates, yet rates were comparable to other studies on inshore turbid reefs, and studies adopting microCT analysis. These lower rates of bioerosion observed across marginal inshore reefs are encouraging as this may facilitate the maintenance of positive reef accretion rates, despite reduced rates of coral carbonate production measured previously on these reefs (2.9 and 3.8 kg $m^{-2}y^{-1}$; See [43]). Macroborers were the dominant drivers of bioerosion at Eva and Fly reefs, which was positively associated with turfing algae, highlighting the important role that fish may play in bioerosion rates on these reef types. Importantly, the proportion of macroboring taxa observed within experimental *Porites* blocks was comparable to that observed within coral rubble samples, increasing our confidence in the representation of the yearly bioerosion rates to longer-term rates. We also saw an inverse relationship between microboring rates and temperature, potentially due to temperature anomalies recorded during the summer months. This may suggest that, at least for micro-borers, bioerosion rates may decline with future rising SSTs, although as more dead substrate following increased coral mortality occurs concurrently, this may offset the decline in micro-boring activity rates. Given the difficulties in accurately assessing in situ rates of endolithic bioerosion, an improved understanding of relationships between environmental drivers, habitat preferences and grazing pressure with endolithic bioeroding communities is needed to improve predictions of reef carbonate loss with future climate change.

**Author Contributions:** S.D. is responsible for the study conception and design. Material preparation was performed by T.D. and S.D., data collection was performed by S.D. and J.N., and analysis was performed by S.D. and I.L. The first draft of the manuscript was written by S.D. and all authors commented on previous versions of the manuscript. All authors have read and agreed to the published version of the manuscript.

**Funding:** This research study was supported by an Australian Research Council (ARC) Discovery Project DP160102561; and a DECRA Fellowship DE180100391 awarded to Nicola Browne of Curtin University, Perth, as part of the island resilience project (2018–2020).

**Institutional Review Board Statement:** Not applicable.

**Data Availability Statement:** The data presented in this study are available on request from the corresponding author. The data are not publicly available due to privacy reasons.

**Acknowledgments:** The authors acknowledge the facilities and scientific and technical assistance of the National Imaging Facility, a National Collaborative Research Infrastructure Strategy (NCRIS) capability, at the Centre for Microscopy, Characterisation and Analysis, The University of Western Australia. We would like to acknowledge and thank the Minderoo Foundation for their assistance in the field. Thank you to Brooke Gibbons, Savita Goldsworthy and Adi Zweifler for assistance in the field.

**Conflicts of Interest:** The authors have no financial or proprietary interests in any material discussed in this article.

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
