# Peer review of "Low Bioerosion Rates on Inshore Turbid Reefs of Western Australia"

_diversity, doi:10.3390/d15010062_

Round 1

Reviewer 1 Report

The manuscript is well written, the study is clear in the definition of objectives and well-performed. The interpretation of results is also well justified and within the appropriate context.  Some of the interpretations could be controversial, but this is exactly what this type of studies need to generate, discussion.  There are a few problems that need to be resolved, particularly concerning to the light measurements, although minor points that can be fixed easily with minor revision. My recommendation is acceptance after these problems are resolved or justified:

(1)   The abstract states that bioerosion on inshore reefs is expected to increase with global climate change. However, in the discussion it is also concluded that temperature anomalies (by the way there is an error writing anomalies in line 517) may explain a negative association between temperature and microboring. Please resolve this apparent inconsistency. How do you expect increases in bioerosion associated with global climate change if global warming is affecting negatively this process?  

(2)   The description of the light environment is very poor. Light is a parameter that present very large diurnal variability (from 0 to the peak of irradiance at noon), as well as among days depending on cloud cover, hydrodynamics, etcc. The use of a single value to describe the differences between two reefs is insufficient, particularly if there is no information about the “comparability” of both descriptions.  How, when and where irradiance values were recorded? At what time of the day? Were similar days, with no clouds, and similar time of the day when both measurements were recorded? The manuscript does not provide enough information to evaluate the accuracy of the light descriptions. Specifically, a single value, instantaneous with no context or as average for a certain period of indetermined time, is meaningless for the characterization of differences in light availability between the different sites investigated.  The optimal description for a better characterization of each light environment, would have been the description of the attenuation coefficient for each site (kd in m-1) or measures of Irradiance simultaneously outside the water column and at a specific depth, comparing then differences in the % of surface irradiance that site receives. Light characterizations require more attention, particularly if light seems to be a key environmental parameter for the interpretation of results.  

(3)   Discussion (lines 481 to 489). The interpretation that the association between micro-borers and macroalgal cover is mediated by grazing pressure is too simplistic.  Under nutrient enrichments macroalgal growth could be high under high grazing pressure, as some alga are very fast growers and could saturate grazing, escaping from its control. Depending on the levels of nutrients, the type of macroalgae and the rate of grazing, macroalgal abundance may not be related to the presence of herbivores.

Reviewer 2 Report

Really well written and organized paper with nice methodology and testable aims, this project involved a lot of work! I don’t have many comments – apart from a few minor changes I think the manuscript is ready to be published.

Specific comments:

Abstract

Lines 19-21: the connection between the two parts of the sentence - worms and turfing algae, and the role of grazing fish - is not clear, how are they linked?

Lines 21-23: Is the reef still showing evidence of positive accretion despite reduced coral cover and carbonate production (and in comparison to what reefs?)

Introduction

1st paragraph: might be good to briefly list other causes of erosion of reefs

Line 39: these reef systems are expanding? Or just under greater pressure?

Line 74: patchily or patchy?

Line 86-87 – “making assuming” à assuming

Nice summary of techniques used to measure endolithic bioerosion.

Should spell out microCT on first use and provide brief definition

Figure 2 needs a, b, c – microCT image doesn’t look like a traditional microscope image – how has this been altered? – scale bars would be useful too

Line 465 – fullstop missing
